# Evaluation of plasma N-terminal pro-B-type natriuretic peptide levels in healthy North American Salukis with normal echocardiographic measurements

Christopher Brennan[☯], Tamilselvam Gunasekaran[iD][☯], Robert A. Sanders[iD]*[☯]

Department of Small Animal Clinical Sciences, College of Veterinary Medicine, East Lansing, Michigan, United States of America

☯ These authors contributed equally to this work.
* ras@msu.edu

**Data Availability Statement:** All relevant data are available on Mendeley Data (doi: 10.17632/gj2p86tp9v.1).

## Abstract

Measurement of N-terminal pro-B-type natriuretic peptide (NT-proBNP) levels has been shown to have clinical significance for diagnosis and management of heart disease in dogs. Evaluation of current reference limits for specific breeds is necessary to ensure the test can accurately distinguish between healthy and diseased animals. The objective of this study is to evaluate the adequacy of currently established NT-proBNP reference limits for clinical use in healthy Salukis. Cardiac health of 33 clinically healthy Salukis was evaluated via echocardiography using available breed standards. Plasma concentrations of NT-proBNP were measured using a commercially available assay. A one-sided 97.5% upper reference limit for the NT-proBNP concentrations was calculated using non-parametric percentile method. The 97.5% upper reference limit was 769 pmol/L (90% CI, 547–1214 pmol/L) for the study dogs. This upper reference limit was within the currently established non-breed specific NT-proBNP upper reference limit of 900 pmol/L. No relationship between sex, age, or body weight on plasma levels of NT-proBNP was noted. Results of this study supports the use of currently available non-breed specific NT-proBNP cut-off values for clinical evaluation of healthy Salukis.

## Introduction

Initial evaluation of cardiac disease in veterinary patients has traditionally lacked a readily accessible and objective method of quantitative evaluation. Methods such as, electrocardiography, thoracic radiography and echocardiography can be inaccessible to clinicians and owners alike. In humans, evaluation of natriuretic peptide levels has been widely used for quantitative assessment of various cardiac diseases. In human patients with heart failure, natriuretic peptides not only have utility as a diagnostic tool but can also be used to develop prognoses and inform treatment strategies [1, 2]. Natriuretic peptides have also been used in the management of structural heart diseases, acute coronary syndromes, and atrial fibrillation in human patients [1–3].

**Funding:** The author(s) received no specific funding for this work.

**Competing interests:** The authors have declared that no competing interests exist.

Recent research has suggested N-terminal pro B-type natriuretic peptide (NT-proBNP), as a useful biomarker for similar quantitative evaluation of various cardiac diseases in dogs [4–18]. One such purpose is differentiating between cardiac and non-cardiac causes of respiratory distress in canine and feline patients [5–10]. Research has also shown value in using NT-proBNP levels for monitoring and prediction of mortality in cases of myxomatous mitral valve degeneration [4, 11–16]. Furthermore, Increased NT-proBNP concentrations have been associated with dilated cardiomyopathy and may have the potential to be used to screen for the disease in Doberman Pinschers [17, 18].

Several veterinary studies have highlighted breed specific differences in echocardiographic measurements especially in sight hounds such as Greyhounds [19, 20]. Greyhounds have unique echocardiographic indices and a higher heart weight to body weight ratio when compared with other breeds; they have also been found to have larger vertebral heart sizes than other breeds [19–23]. Changes in echocardiographic variables appear to persist even in non-racing Greyhounds [20]. Greyhounds also have several biochemical analytes that differ significantly from other breeds, including higher than average cardiac troponin levels and plasma NT-proBNP levels in healthy, retired racing dogs [23–25]. Much like Greyhounds, normal Salukis have also been found to have unique echocardiographic measurements that are different from other breeds [21, 26]. Given the variability among breeds as a whole and given that a related breed with similarly unique echocardiographic indices appears to have significantly increased levels of plasma NT-proBNP, there is merit in evaluating circulating NT-proBNP in the Saluki dogs.

The goal of this study was to evaluate currently available NT-proBNP assay reference limits for use in healthy Saluki dogs with normal echocardiographic measurements. It was hypothesized that Salukis would have elevated NT-proBNP concentrations when compared to currently available reference cutoffs.

## Materials and methods

### Animals

Animals were included in the study if they were clinically healthy and had echocardiographically normal measurements established for purebred Salukis [21, 26]. Echocardiographic examinations were performed as previously described and results were compared to breed-specific normal ranges [21, 26]. All animals were evaluated at one of two Saluki Club of America National Specialty Shows. The study protocol was approved by the Institutional Animal Care and Use Committee (PROTO201800022) at Michigan State University (PROTO201800022). Written informed consent authorizing study participation was obtained from participating owners. No sedation or anesthesia was performed for sample collection in this study.

### Blood sample collection and NT-pro BNP analysis

Blood was sampled via venipuncture of external jugular vein, collected in 5 mL EDTA tubes, centrifuged at 2500 RPM, and supernatant stored in a freezer at -20° C. Samples were shipped in two separate batches over dry ice to a commercial laboratory for analysis. Samples were analyzed using a commercially available second-generation ELISA test (Canine Cardiopet® proBNP test kit, IDEXX Laboratories Inc., Westbrook, ME.). The test has been previously validated for use in dogs [27].

### Statistical methods

Statistical analysis was performed using commercially available software (MATLAB, Version 9.8 (R2020a), Natick, Massachusetts: The MathWorks Inc.). Distribution of NT-pro BNP

concentrations was assessed for normality using the Shapiro-Wilk test. Due to non-normal distribution, plasma NT-proBNP concentrations were presented using medians and interquartile ranges. The Mann-Whitney U test was used to compare NT-proBNP concentrations between sexes. Simple linear regression analysis was performed to assess for association between age and serum concentrations of NT-proBNP. The 97.5% upper reference limit was estimated by using a bootstrap method (The dataset was iteratively sampled 10,000 times with replacement). A 90% confidence interval around the upper reference limit was constructed in an identical fashion.

## Results

Forty-three dogs were initially evaluated, with 33 dogs included in the final analysis. Ten dogs were excluded for abnormalities noted during the echocardiographic exam, including mitral regurgitation, aortic regurgitation and tricuspid regurgitation. Eighteen of the dogs were female and fifteen were male. The median age was 54 months (IQR = 43 mos, P25 = 26.5 mos, P50 = 69.3 mos). The median weight of the dogs was 21.9 kg (IQR = 2.9 kg, P25 = 20.3kg, P50 = 23.2kg).

The median plasma NT-proBNP concentration was 250 pmol/L (IQR = 93.5 pmol/L, P25 = 250 pmol/L, P75 = 250 pmol/L, see **Fig 1**). The majority of samples (24 out of 33, 72.7%) measured at the lower limit for detection for the assay used in this study (i.e. 250 pmol/L). The 97.5th percentile (upper reference limit) was 769 pmol/L (90% Confidence Interval, 547–1214 pmol/L). There was no significant difference in plasma NT-proBNP concentrations between male and female dogs (p = 0.9). There was no significant correlation between age and plasma NT-proBNP concentrations (r = 0.19, p = 0.29) nor between body weight and plasma NT-proBNP concentrations (r = -0.19, p = 0.29).

## Discussion

Measurement of plasma NT-proBNP concentrations presents a potentially useful screening tool when evaluating the dogs for heart disease. Current commercially available tests for use in the general canine population utilize a reference interval constructed from many dogs of various breeds [5]. However, considerable variability in plasma NT-proBNP concentrations has been noted between breeds [28], suggesting merit in evaluating NT-proBNP concentrations on a breed-by-breed basis.

To our knowledge, plasma levels of NT-proBNP have not been previously evaluated in Salukis. This study intended to evaluate NT-proBNP concentrations in apparently healthy Salukis in comparison with currently used non-breed specific reference ranges which have an upper cut-off of 900 pmol/L. The results of this study support use of a cutoff of 900 pmol/L in Salukis, as this value falls within the 90% confidence interval of the calculated 95th percentile.

The current study has several limitations including small sample size. Furthermore, while best efforts were made to rule out other systemic illness, no clinicopathologic diagnostics were used to make these decisions, nor was follow up performed on any of the dogs. It is therefore possible that animals with higher NT-proBNP levels had occult cardiac or other systemic illness that was not apparent on physical examination.

Finally, the test used in the current study cannot detect serum NT-proBNP concentrations below 250 pmol/L, and a large proportion of samples returned measuring at this low cutoff value. As a result, the specific concentrations of NT-proBNP in many of the collected samples remain unknown, as their true values could be anywhere from 250pmol/L or below. This does not prove to be an issue in a clinical setting, as a low NT-proBNP concentration is not of any clinical significance. However, in the setting of this study, the lack of sample diversity

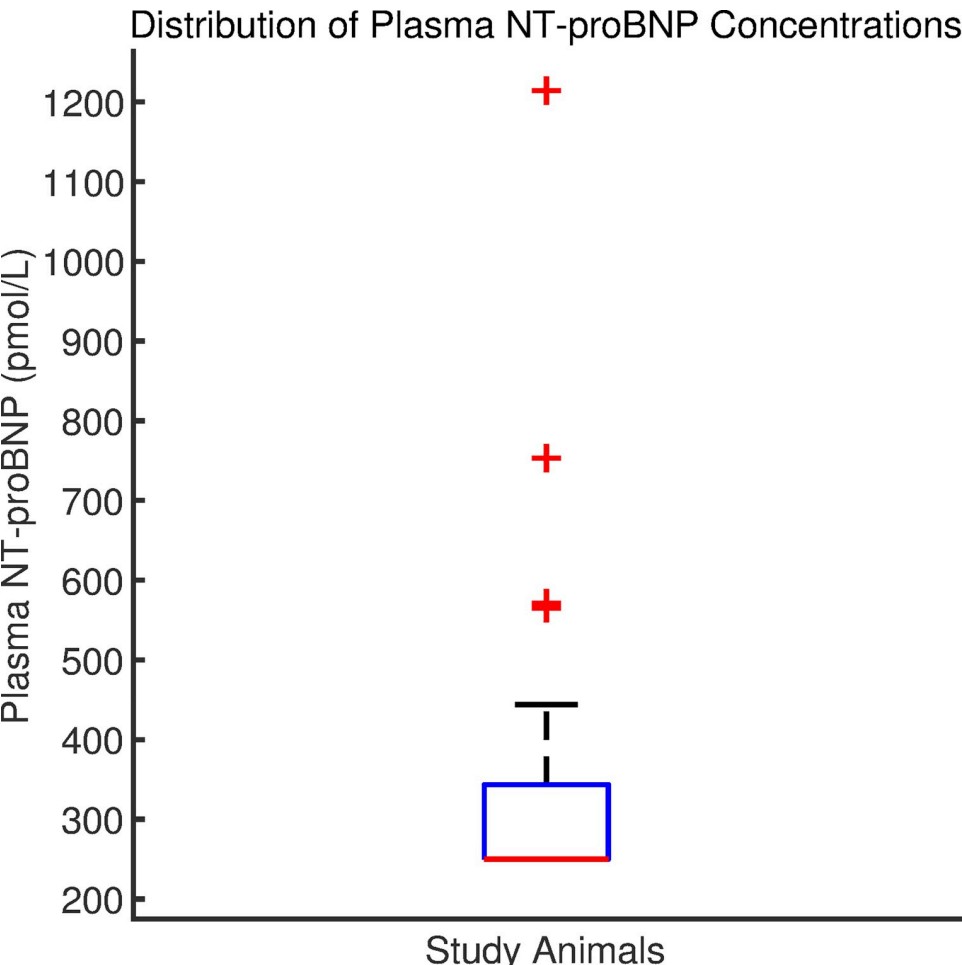

**Fig 1. Boxplot of NT-proBNP concentrations in 33 healthy Salukis.** The red bottom line denotes both the median value and lower quartile (250pmol/L), while the top of the box denotes the upper quartile. The black whisker marks the highest value that is not an outlier, and values beyond this upper bound are marked with plus signs (+).

precludes generation of a true reference interval, as the recommended logarithmic transformation and robust evaluation of the data is impossible [29].

Interestingly, the 97.5% percentile upper limit calculated in this study is lower than the assay upper limit of 900 mmol/L. This would be an important distinction when screening for cardiac disease in the breed. Say, for example, if a true reference interval is generated with an upper limit that is significantly lower than 900 pmol/L. In this scenario, an increased NT-proBNP level in a diseased Saluki could be misinterpreted as a normal concentration when using the standard reference limit. As such, more work is required to determine a true breed-specific NT-proBNP reference range.

## Conclusion

Most healthy Saluki dogs have NT-proBNP concentrations that confirm to the currently established upper reference limit for the commercial assay. However, further research is needed to evaluate the adequacy of NT-proBNP upper reference limits in differentiating Saluki's with and without cardiac disease by establishing true reference limits.

## Author Contributions

**Conceptualization:** Robert A. Sanders.

**Data curation:** Tamilselvam Gunasekaran.

**Formal analysis:** Christopher Brennan, Tamilselvam Gunasekaran.

**Investigation:** Robert A. Sanders.

**Methodology:** Robert A. Sanders.

**Project administration:** Robert A. Sanders.

**Supervision:** Robert A. Sanders.

**Writing – original draft:** Christopher Brennan, Tamilselvam Gunasekaran, Robert A. Sanders.

**Writing – review & editing:** Christopher Brennan, Tamilselvam Gunasekaran, Robert A. Sanders.

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
