## [Decision Letter · Decision Letter 0]

10 Jan 2022

Evaluation of plasma N-terminal pro-B-type natriuretic peptide levels in healthy North American Salukis with normal echocardiographic measurements.

PONE-D-21-34829

Dear Dr. Sanders,

We’re pleased to inform you that your manuscript has been judged scientifically suitable for publication and will be formally accepted for publication once it meets all outstanding technical requirements.

Kind regards,

Vincenzo Lionetti, M.D., PhD

Academic Editor

PLOS ONE

1. Please note that PLOS ONE uses a single-blind peer review procedure. We would therefore be grateful if you could include in the information that has been redacted (line 86) for peer review in the manuscript.

Reviewers' comments:

Reviewer's Responses to Questions

**Comments to the Author**

1. Is the manuscript technically sound, and do the data support the conclusions?

Reviewer #1: Yes

Reviewer #2: Yes

2. Has the statistical analysis been performed appropriately and rigorously? 

Reviewer #1: Yes

Reviewer #2: Yes

3. Have the authors made all data underlying the findings in their manuscript fully available?

Reviewer #1: Yes

Reviewer #2: Yes

4. Is the manuscript presented in an intelligible fashion and written in standard English?

Reviewer #1: Yes

Reviewer #2: Yes

5. Review Comments to the Author

Reviewer #1: This is an interesting article. I think that the results reported in this article can aid to accurately distinguish between healthy and diseased animals. This study has some limitations, as Authors have correctly reported in the Discussion section of the manuscript.

Reviewer #2: The article is well written and the statistical analysis adequate, even if the number of cases is limited in order to accurately determine the reference ranges of NT-proBNP within the breed. It would have been interesting to evaluate this parameter within some greyhound breeds with very different weights and sizes, for example the small Italian greyhound and the Afghan hound.

6. PLOS authors have the option to publish the peer review history of their article (what does this mean?). If published, this will include your full peer review and any attached files.

Reviewer #1: No

Reviewer #2: No

---

## [Editor Report · Acceptance letter]

21 Jan 2022

PONE-D-21-34829 

Evaluation of plasma N-terminal pro-B-type natriuretic peptide levels in healthy North American Salukis with normal echocardiographic measurements. 

Dear Dr. Sanders:

I'm pleased to inform you that your manuscript has been deemed suitable for publication in PLOS ONE. Congratulations! Your manuscript is now with our production department. 

Kind regards, 

on behalf of

Prof. Vincenzo Lionetti 

Academic Editor

PLOS ONE